# Coating Cellulosic Material with Ag Nanowires to Fabricate Wearable IR-Reflective Device for Personal Thermal Management: The Role of Coating Method and Loading Level

**DOI:** 10.3390/molecules26123570

**Published:** 2021-06-11

**Authors:** Mohsen Gorji, Saeedeh Mazinani, Abdol-Rahim Faramarzi, Saeedeh Ghadimi, Mohammadreza Kalaee, Ali Sadeghianmaryan, Lee D. Wilson

**Affiliations:** 1New Technologies Research Center (NTRC), Amirkabir University of Technology, Tehran 1591634311, Iran; s.mazinani@aut.ac.ir (S.M.); rahimfaramarzi@gmail.com (A.-R.F.); s.ghadimi94@aut.ac.ir (S.G.); 2Department of Polymer Engineering, Faculty of Engineering, South Tehran Branch, Islamic Azad University, P.O. Box 19585-466, Tehran 1777613651, Iran; mr_kalaee@azad.ac.ir; 3Nanotechnology Research Center, South Tehran Branch, Islamic Azad University, Tehran 1584743311, Iran; 4Department of Chemistry, Ardabil Branch, Islamic Azad University, Ardabil 5615731567, Iran; sadeghian469@gmail.com; 5Department of Chemistry, University of Saskatchewan, 110 Science Place, Room 165 Thorvaldson Bldg., Saskatoon, SK S7N 5C9, Canada

**Keywords:** personal thermal management, IR-reflector, wound dressing, silver network

## Abstract

Textiles coated with silver nanowires (AgNWs) are effective at suppressing radiative heat loss without sacrificing breathability. Many reports present the applicability of AgNWs as IR-reflective wearable textiles, where such studies partially evaluate the parameters for practical usage for large-scale production. In this study, the effect of the two industrial coating methods and the loading value of AgNWs on the performance of AgNWs-coated fabric (AgNWs-CF) is reported. The AgNWs were synthesized by the polyol process and applied onto the surface of cotton fabric using either dip- or spray-coating methods with variable loading levels of AgNWs. X-ray diffraction, scanning electron microscopy (SEM), infrared (IR) reflectance, water vapor permeability (WVP), and electrical resistance properties were characterized. The results report the successful synthesis of AgNWs with a 30 μm length. The results also show that the spray coating method has a better performance for reflecting the IR radiation to the body, which increases with a greater loading level of the AgNWs. The antibacterial results show a good inhibition zone for cotton fabric coated by both methods, where the spray-coated fabric has a better performance overall. The results also show the coated fabric with AgNWs maintains the level of fabric breathability similar to control samples. AgNWs-CFs have potential utility for cold weather protective clothing in which heat dissipation is attenuated, along with applications such as wound dressing materials that provide antibacterial protection.

## 1. Introduction

In recent years, personal thermal management (PTM) has attracted increasing attention as a cost-effective method for saving energy. In this scenario, the ventilation is localized on the human body instead of the whole surrounding environment [1]. Nanophotonic textiles, including human IR-reflectance textiles [2], human IR-transparent textiles [3], and sunlight IR-absorbent or -reflector materials [1] have led to proposed technologies for achieving PTM clothing. More than 50% of the heat generated by the human body is dissipated through infrared (IR) radiation [2]. IR-reflector textiles reflect the majority of the human body IR radiation and provide warmth for the body in an environment with a lower thermal set point [4]. For this technology, the high emissivity of common fabric (0.9–0.75) provides little radiative insulation effect and is reduced by metals as a class of good infrared (IR) reflectors in the form of nanoparticles [5], nanowires [2], and porous films [6] to afford lightweight and breathable clothing materials.

Among the various types of metallic nanostructures, silver nanowires (AgNWs) are a superb choice because of the high IR reflection efficiency, the high electrical conductivity, potent antibacterial properties, and excellent UV-blocking ability [7,8]. There are also other composite materials, such as a polypyrrole/carbon black coatings that increase electrical connectively of cotton material. Coated cotton with this composite material was reported to be mechanically stable [9]. The high electrical conductivity of AgNWs makes this one-dimensional nanostructure a good candidate for electromagnetic interface shielding and joule heating applications [10]. The antibacterial properties of AgNWs has garnered continued interest because of the minimum agglomeration when the nanowires are fabricated into a film [11].

Various coating methods have been reported for AgNWs that include dip-coating [7,8], thermal evaporation [6], electrodeless plating [2], and vacuum filtering [5]. Apart from the dip-coating, other methods are not suitable for such large-scale production in the textile industry. Depositing AgNWs using the dip-coat method yields a connected network across the thickness of the fabric. This connective wire network is conductive and results in greater thermal conductivity of a coated fabric in contrast with untreated fabric. The presence of AgNWs network attenuates heat dissipation by a radiative modality but simultaneously accelerates the body heat transfer by a conductive mode of action. Ag nanostructure networks may be constructed where one side of a fabric can be coated asymmetrically using a spray-coating method as an industrial-level technology [12]. Spray coating is a conventional method that is facile, rapid, and inexpensive [13,14]. Spray coating has not been widely reported for the AgNWs despite the valuable progress in this field. Thus, further studies directed at large-scale production for practical industrial applications are needed for future technology development [15].

Herein, the effects of two coating methods and the functional properties of the treated fabric are investigated. Highly conductive AgNWs were synthesized and coated onto a cotton fabric surface using dip-coating and spray-coating methods, respectively. The resulting AgNWs and coated fabrics with AgNWs (AgNWS-CF) were characterized by X-ray diffraction, scanning electron microscopy, thermal camera imaging, water vapor permeability, electrical conductivity, antimicrobial, and washing tests. The loading and coating method for AgNWs, along with the effects of such coatings onto cotton fabrics, were investigated in detail, where it is anticipated that this study will contribute to the valorization of such AgNWs-CF materials.

Fabrics serve as the main structural design for various textile products include sport and medical wound dressings, and protective clothing. Such textiles often demand multifunctional properties that are lightweight, breathable, warm, antibacterial, and UV-blocking [4,6]. Nanostructured materials such as silver, clay, graphene, graphene oxide, and metal oxides (e.g., TiO_2_, ZnO) impart desired functional properties to the fabrics. This includes ultraviolet protection, antibacterial resistance, flame retardancy, antistatic, electrical conductivity, self-cleaning, and super hydrophobicity [16,17]. Multifunctional fabrics with integrated functionalities that incorporate nanomaterials offer synergetic benefits that extend beyond the sum of the individual material properties. Reducing the size, weight, cost, power consumption, and complexity of such nanocomposites, along with improving the efficiency, safety, and versatility are advantageous for the integration of nanomaterials [7].

## 2. Results and Discussion

### 2.1. Morphology and Structural Characterization

Figure 1 shows the XRD patterns for samples of AgNWs. The diffraction peaks at 38.21°, 44.39°, 65.54°, and 77.49° indicate the crystal planes (111), (200), (220), and (311) of the AgNWs with a face-centered-cubic (FCC) crystal structure [18].

The microstructure and morphology of AgNWs and AgNWs-CF play a major role in their IR reflection properties. Thus, the morphologies of AgNWs and AgNWs-CF are investigated by scanning electron microscopy (SEM), as shown in Figure 2. According to the SEM results in Figure 2, the synthesized nanowires have a regularly morphology with a length of around 30 μm, which concur with the results from a published study [19]. Moreover, the distance between the nanowires must be less than the thermal wavelength of the body (with the dominant peak of 9 microns) in order for the structure to reflect heat waves efficiently. The SEM images reveal that the nanowire network meets the desired size dimensionality (cf. Figure 2).

In Figure 2c, the bimodal diameter of cotton fiber with around 10 μm and densely packed metallic nanosilver network with below 500 nm are presented. Figure 2 shows not only the silver nanowire engages with fiber surface but also forms a network between adjacent fibers and yarn. This network decreases the pore size from upper limit of 100 μm (blank cotton fabric) to around 100 nm (AgNWs-coated cotton fabric). Here, the network of AgNWs can provide good water vapor permeability required for thermal comfort, while it can successfully reflect IR-radiation from the human body [5]. This decrease in pore size is also useful for personal protective equipment (PPE), such as textiles with antiviral and anticoronavirus properties, where the particle size of the pathogens are approximately 120 nm [20].

### 2.2. Thermal Management Characterization

Figure 3 shows the thermal image of samples on the human hand in the start (~5 s), middle (~150 s), and at the end (~300 s) of the test for blank and coated fabrics. For each sample code, the mean temperature of coated samples (t1), the mean temperature of blank samples (t2), and the temperature difference between coated and blank samples (∆t) are presented. The fabric effectively reflects human body radiation, because the pore size created by the nanowire network is below the wavelength range of body radiation with a dominant peak of 9 μm.

With an increased loading of AgNWs, a greater thermal difference indicates greater reflectance of human body radiation. The maximum decrease in temperature is near 3.7 °C for spray-coated fabric, as compared with the blank sample, in agreement with other reported studies [2,6]. This indicates that silver coated fabric is an effective IR reflector that reduces heat loss from the human body, and the expansion of a heating set point can lead to space heating energy savings up to 25% [2]. A comparison between the dip- and spray-coating methods indicate that greater material performance was obtained from the spray-coating method. This may relate to a relatively uniform surface coverage of the AgNWs deposited by this coating method.

Figure 4 shows the temperature difference (ΔT) for each sample, including coating and spray methods, as compared to the blank sample at the start, middle, and end times of the test period. Samples treated by the spray method showed a larger ΔT than the control samples (blank) by the coating method. In other words, at coating level 1, the treated sample in spray performed better than the treated sample in the coating method, and the same is true at coating levels 2 and 3. In both cases, with increasing the number of nanowires, the temperature difference between the coated samples is more significant than the control sample.

### 2.3. Water Vapor Permeability

One of the key factors in choosing thermal insulation material for use in clothing materials is their breathability. A good insulation layer should prevent loss of body generated heat transfer from the system (human body) to the external surroundings while having a minimal decrease in the breathability of the insulation material. Here, we consider WVP as a breathability index [3]. Figure 5 shows the water vapor permeability for the blank sample (normal cloth) and coated fabric materials prepared by the dip- and spray-coating methods. Although the breathability of coated samples is slightly less than the blank sample, there is no significant difference between them. In other words, AgNWs coating does not appear to affect the breathability and thermal comfort of coated samples. This result may be attributed to the relative size of water molecules (2.7 angstroms) and the relative pore size of nanowire coating (>100 nm), which is estimated to permit water molecules to pass easily, if one neglects the role of chemical interactions with the material. This result is in accordance with the result of Hsu et al. [8], where a small decrease in the WVP of samples coated with AgNWs was reported.

### 2.4. Electrical Conductivity and Washing Fastness

The high electrical conductivity of silver nanowires allows the coated fabric to be potentially used as a suitable material for heat conduction [6]. As Figure 6 shows, the electrical resistance of the coated fabric is about 1 Ω and within the range of reported results [8,21,22]. This low electrical resistance makes AgNWs-CF a promising candidate for the joule heating mechanism [23].

Washing fastness of coated fabric was measured to identify the durability of the silver nanowires. As Figure 6 shows, the coated fabrics retain good electrical conductivity in the repeated washing cycles. The increased electrical resistance of the spray-coated fabric in successive washing can be attributed to greater entanglement of the nanowires and the cotton fibers, in agreement with the morphology of the material prepared by the dip coating method.

### 2.5. Antibacterial Testing

The antibacterial properties of the AgNWs-CF were evaluated using Gram-negative *E. coli* and Gram-positive *S. aureus*, which are two typical and relevant bacteria (Figure 7) that affect human health.

As shown in Figure 7 (1–5), an empty gap around the AgNWs-CF prepared by both coating methods reveals the presence of an inhibition zone. The AgNWs reveal the capability to overcome the growth of bacteria before and after the five-cycle washing test. These results indicate an improved antibacterial performance of the spray-coating method. The release of the Ag ions from Ag nanowires under such wet conditions in the agar plate system enable interaction of the Ag species with the bacterial cell membrane, resulting in loss of DNA replication function and inactivation of the bacterial proteins [5]. One of the interesting results is the different effect of AgNWs in contact with *E. coli* and *S. oreus* reveals that the effect is weaker for the AgNWs-CF for preventing *S. oreus* replication. The differential antibacterial activity between *E. coli* and *S. oreus* may be attributed to differences in the shape and structure of these two Gram-positive and Gram-negative bacteria [24].

## 3. Materials and Methods

### 3.1. Materials

Poly(vinylpyrrolidone) (PVP) is a polymer with a molecular weight of 360,000 g/mol. Ethylene glycol, silver nitrate (AgNO_3_), iron (III) chloride (FeCl_3_), ethanol, and acetone were purchased from Merck and used without further purification. Cotton fabric was purchased from Yazdbaf Co., Yazd, Iran.

### 3.2. Silver Nanowires Synthesis

The silver nanowires were synthesized using a polyol process, as described by Jiu et al. [19]. For this purpose, 0.2 g of PVP with a molecular weight of 360,000 was first added to 25 mL of ethylene glycol and completely dissolved at room temperature with a magnetic stirrer. Then, AgNO_3_ (0.25 g) was added to the PVP solution. Upon complete dissolution, a clear and uniform solution is obtained. Finally, 3.5 g of a salt solution of iron chloride (600 μM in ethylene glycol) was added to the mixture at once, and the resulting mixture was stirred for two minutes. The mixture was then immediately transferred to a reactor that reached 130 °C to grow nanowires for 5 h. In the next step, the product was washed using acetone and ethanol, followed by centrifuging at 4000 rpm for 5 min. Finally, the AgNWs were dispersed into ethanol for applying onto the cotton fabric substrate.

### 3.3. Preparation of Nanowire Coated Fabric

Cotton fabric has wide utility in clothing since it serves as a good substrate for wound dressing materials. For obtaining different loadings of AgNWs, variable concentrations of AgNWs were prepared in the mixture of ethanol and acrylic resin. In the dip-coating method, the fabric samples were immersed in the mixture for 5 min, whereas the spray-coating method involves spraying of the mixture onto a cotton fabric. The process was followed by drying at 90 °C for 30 min. Table 1 provides a summary of the prepared samples according to the two methods of application.

### 3.4. Materials Characterization

X-ray diffraction (XRD) profiles of the synthesized AgNWs were obtained using a Phillips PW3040-MPD diffractometer (Royal Philips Corp., Amsterdam, Netherlands). Morphological characterization of the AgNWs and AgNWs-CF was done using scanning electron microscopy (JSM-7800F FESEM, JEOL Ltd., Tokyo, Japan).

The IR-reflecting properties of the samples were measured by a thermal camera device. The samples were cut in dimensions of 4 cm × 4 cm, where the blank and coated samples were located simultaneously in the palm of the experimentalist’s hand. The IR-transient behavior was measured for 5 min, where a mean of 10 points was reported for each sample.

The water vapor permeability (WVP) test was carried out based on an upward cup method, according to the ASTM E96 method. A 100 mL cup filled with 50 mL distilled water was sealed by the fabric samples and gaskets. Then, the sealed cup was placed into a chamber with temperature and relative humidity of 28 °C and 40 ± 10%, respectively. The total mass of the cup was measured, and the reduced mass, corresponding to the evaporated water, was then divided by the exposed area to derive an estimate of the WVP.

A Sa-Iran digital multimeter (model 156 8515, Iran Electronics Industries, Tehran, Iran was used for electrical surface resistivity of the coated samples using a two-probe method with a probe distance of 1 cm.

The antibacterial activity test was done with Escherichia coli (*E. coli*) and Staphylococcus aureus (*S. aureus*) bacteria, which were developed in the LB broth medium. Using an autoclave, the culture media and samples were sterilized at 121 °C for 3 h. Then, 80 μL of diluted *E. coli* or *S. aureus* suspensions were layered onto LB agar plates. The obtained AgNWs-CF, before and after washing tests, were placed in the middle of the LB agar plates and pressed gently. The test was done at 37 ± 1 °C for 24 h. The size of the inhibitory zone was considered as the metric for assessing the antibacterial activity of coated samples.

The mean data collected were shown from the difference between the blank sample and other samples and analyzed by one-way ANOVA using Tukey’s test (95% confidence intervals) by SSPSS V.18. A *p*-value < 0.05 was considered statistically significant.

## 4. Conclusions

In this study, the effects of two conventional coating methods on the performance of AgNWs-CF are presented. This study shows that the spray method provides better IR-reflectance and antibacterial properties relative to the dip-coating method. IR-reflectance of fabric coated by both methods increases with greater loading levels of AgNWs. Antibacterial tests revealed that the AgNWs-CF prepared by both dip- and spray-coating methods can yield inhibition of bacterial growth to at least five washing test cycles. The fabricated structures also showed low electrical resistance before and after five cycles of washing. The observation of favorable IR-reflectance, antibacterial properties, and breathability of the AgNWs-CF provides complementary evidence of suitable properties for such a functional textile material. In particular, the relative pore size of ca. 100 nm for such coated fabrics with AgNWs favors the use of such coated fabrics for cold weather protective clothing and wound dressing, along with potential utility in PPE clothing and facemask materials for protection against coronavirus transmission.

## Figures and Tables

**Figure 1 molecules-26-03570-f001:**
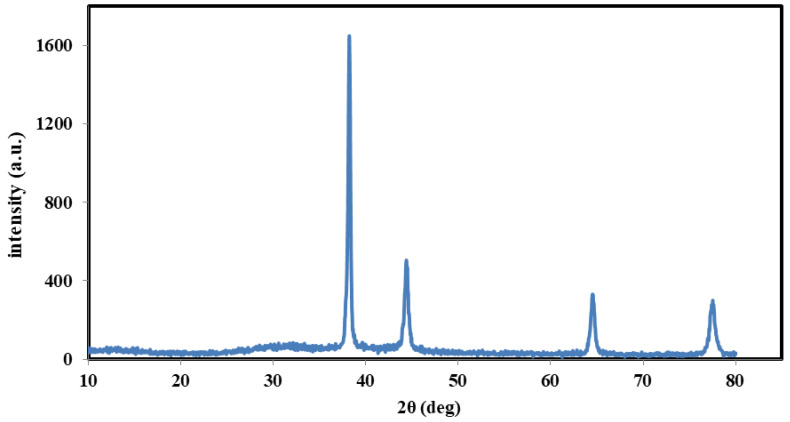
XRD diagram showing the crystal growth planes of prepared silver nanowires.

**Figure 2 molecules-26-03570-f002:**
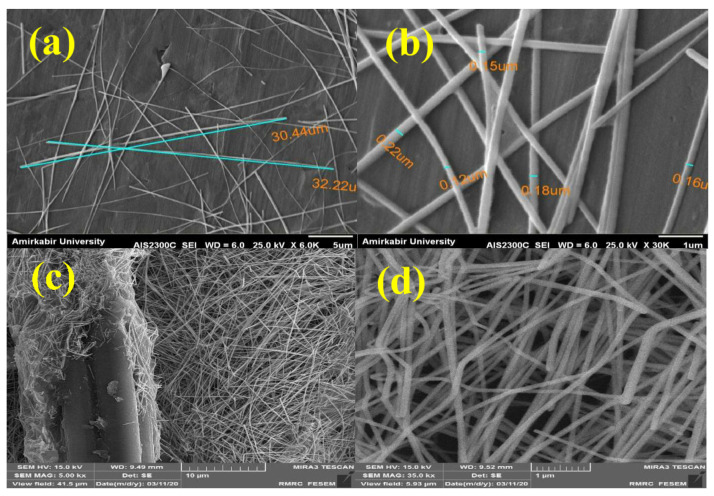
SEM images of the silver nanowires in different magnifications (**a**,**b**), SEM images of cotton fabric coated with silver nanowires (**c**,**d**).

**Figure 3 molecules-26-03570-f003:**
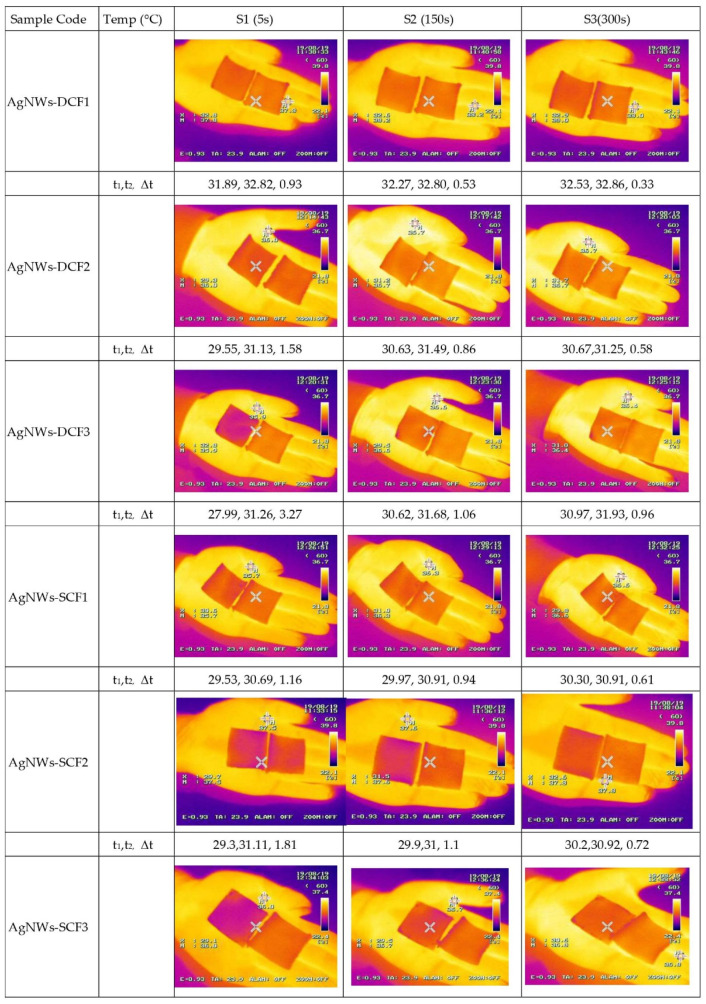
Thermal camera results of samples measured at different time intervals (~5 s, ~150 s, and ~300 s). In the left column, the sample number, mean temperature of coated samples, and mean temperature of the blank sample are presented for variable time intervals.

**Figure 4 molecules-26-03570-f004:**
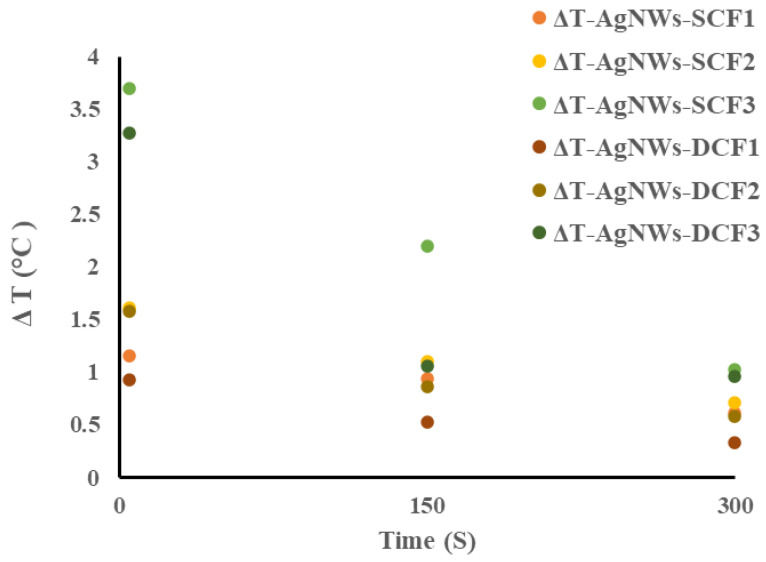
The temperature difference (ΔT) for each sample, including immersion and spray methods, as compared with the blank sample at the start, middle, and end times of the test interval.

**Figure 5 molecules-26-03570-f005:**
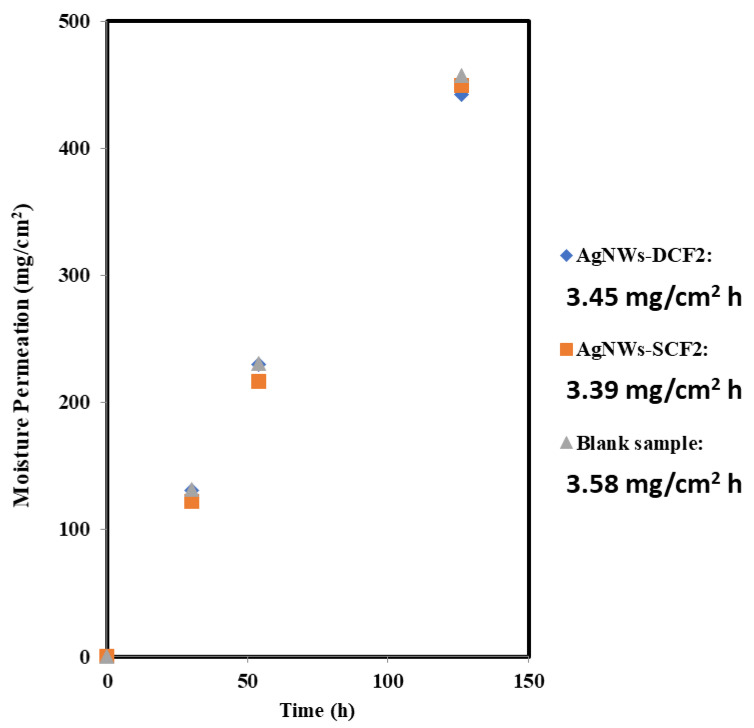
Moisture permeation of samples vs. time for blank and coated sample with dip- and spray-coating methods.

**Figure 6 molecules-26-03570-f006:**
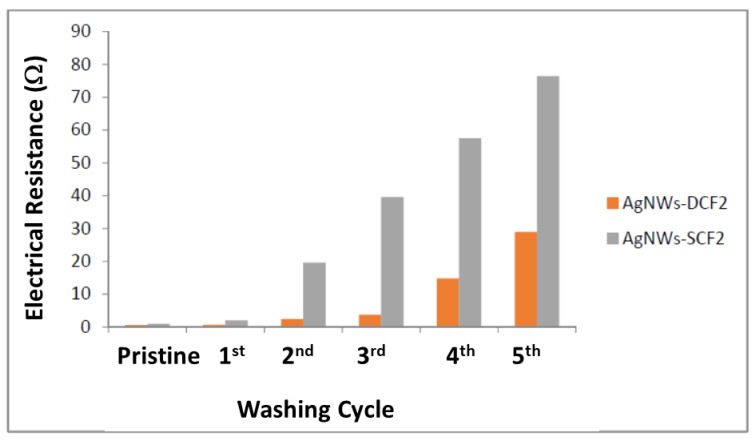
The electrical resistance of the AgNWs coated fabric by dip and spray coating method, before and after cyclic washing for up to 5 cycles of washing.

**Figure 7 molecules-26-03570-f007:**
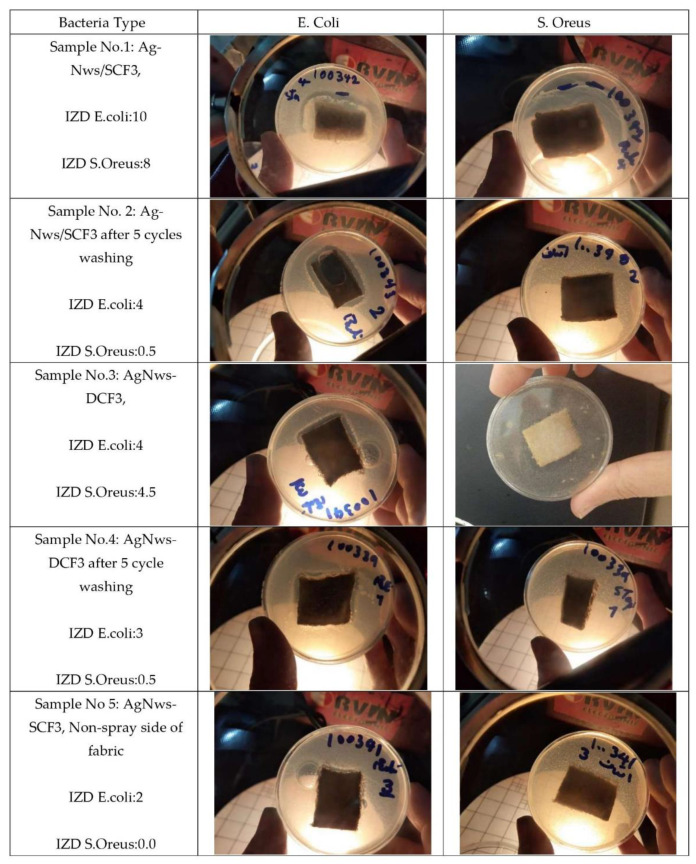
Antibacterial test of samples (1) AgNWs-SCF3, (2) AgNWs-SCF3 after 5 cycles of washing, (3) AgNws-DCF3, (4) AgNws-DCF3 after 5 cycles of washing, (5) AgNWs-SCF3 (non-spray side of fabric). IZD: Inhibition zone diameter.

**Table 1 molecules-26-03570-t001:** Prepared samples with dip- and spray-coating methods.

Sample Code	Loading Method	Loaded Material	AgNWs Loading
AgNWs-DCF1	Dip	AgNWs + acrylic resin	0.1 mg/cm^2^
AgNWs-DCF2	Dip	AgNWs + acrylic resin	0.3 mg/cm^2^
AgNWs-DCF3	Dip	AgNWs + acrylic resin	0.5 mg/cm^2^
AgNWs-SCF1	Spray	AgNWs + acrylic resin	0.1 mg/cm^2^
AgNWs-SCF2	Spray	AgNWs + acrylic resin	0.3 mg/cm^2^
AgNWs-SCF3	Spray	AgNWs + acrylic resin	0.5 mg/cm^2^

## Data Availability

The data presented in this study are available in this article.

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
