# Peer review of "Coating Cellulosic Material with Ag Nanowires to Fabricate Wearable IR-Reflective Device for Personal Thermal Management: The Role of Coating Method and Loading Level"

_molecules, 2021, doi:10.3390/molecules26123570_

Round 1

Reviewer 1 Report

L 70-71 Reference [11] does not describe spray coating resulting in AgNW coating, but Ag clusters.

L 160-233 and Much better characterization of the materialc can be obtained using IR spectrometer equipped with an integrating sphere and calculation of eT out of the spectra than with the thermal main benefit of such coating on textiles.

Please use consistent abbreviations (AgNW vs., Ag NW vs. Ag-NW, AgNW/CF vs AgNW-CF, AgNWs vs. AgNWS)

camera. The spectroscopy method would allow also for precise determination of ageing effect and even some explanation of ageing mechanisms.

L 280-285 I miss IR reflectivity assessment after washing here as I understand heat management as the main feature of the coating.

Author Response

Authors’ Response to Reviewer Queries on MS ID - molecules-1236499

Reviewer #1

L 70-71 Reference [11] does not describe spray coating resulting in AgNW coating, but Ag clusters.

Response:

The important point in the above noted sentence is that the spraying method can be used as a common and easy method to coat fabrics with silver nanostructures. The sentence has been rewritten accordingly.

L 160-233 and Much better characterization of the materialc can be obtained using IR spectrometer equipped with an integrating sphere and calculation of eT out of the spectra than with the thermal main benefit of such coating on textiles.

Response:

The referee's comment is insightful and the suggested method is also possible. However, the use of thermal cameras was more feasible due to its accessibility; hence, the use of thermal camera imaging was used to gain insight on the thermal management properties (cf. Figs. 3-4).

Please use consistent abbreviations (AgNW vs., Ag NW vs. Ag-NW, AgNW/CF vs AgNW-CF, AgNWs vs. AgNWS)

Response:

The use of abbreviations has been corrected and harmonized.

camera. The spectroscopy method would allow also for precise determination of ageing effect and even some explanation of ageing mechanisms.

Response:

This comment is not clear. The stability of the AgNWs to cyclic washing is reported upon in Fig. 6, revealing that

L 280-285 I miss IR reflectivity assessment after washing here as I understand heat management as the main feature of the coating.

Response:

The electrical conductivity test was used to determine the washing stability. Before and after washing, the electrical conductivity of the sample was studied and there was no significant difference, so it can be concluded that in washing, all properties, including IR reflectivity, are preserved. Hsu et al. [9] as a pioneer group of the concept of IR reflector Metallic Nanowire-Coated Textile also evaluated electrical conductivity before and after washing test for washing fastness of nanostructures. 

In summary, the authors acknowledge the insightful and constructive comments of Reviewer #1. The corresponding changes are reflected in the revised manuscript. The authors have also comprehensively edited the manuscript for language, clarity, and syntax throughout to meet the high standards of this journal.

Reviewer 2 Report

Executive Summary

  • The manuscript titled “Coating Cellulosic Material with Ag Nanowires to Fabricate Wearable IR-Reflective Device for Personal Thermal Management: The Role of Coating Method and Loading Level” is a pilot study to investigate the potential use of a silver coating to provide thermal management. Overall, the manuscript is well written with good results. However, when presenting results, authors may need further discussion to provide support for the conclusion. Therefore, major revisions are recommended.

Major Comments

  • 2.2 Thermal Management Characterization and Figure 3.
    • “The results also show that the spray coating method has a better performance for reflecting the IR spectrum to the body that increases with a greater loading level of AgNWs” is not supported by section 2.2 and Figure 3. The easiest way to prove that is to draw a line chart, in which the temperature improves as time goes. Further, samples with greater loading have a higher slope of temperature elevation. Based on my observation of data, this is a feasible solution.
  • Please explain why results were provided selectively.
    • In figure 4, only AgNWs-DCF2 and AgNWs-SCF2 were used.
    • In figure 5, only AgNWs-DCF2 and AgNWs-SCF2 were used.

Minor Comments

  • Please ensure that all figures meet quality requirements as I can barely see the writing.
  • Line 241, “…blank sample there is no significant difference between them.
    • Significant differences must be supported through data analysis. A P-value needs to be provided.
  • Line 306: “As shown in Figure 4 (1-5),…”.
    • I think the authors are referring to Figure 6.
  • Table 1. Prepared samples with dip and spray methods
    • There are three “AgNWs-SCF1”

Author Response

Reviewer #2

Executive Summary

  • The manuscript titled “Coating Cellulosic Material with Ag Nanowires to Fabricate Wearable IR-Reflective Device for Personal Thermal Management: The Role of Coating Method and Loading Level” is a pilot study to investigate the potential use of a silver coating to provide thermal management. Overall, the manuscript is well written with good results. However, when presenting results, authors may need further discussion to provide support for the conclusion. Therefore, major revisions are recommended.

Major Comments

  • 2.2 Thermal Management Characterization and Figure 3.
    • “The results also show that the spray coating method has a better performance for reflecting the IR spectrum to the body that increases with a greater loading level of AgNWs” is not supported by section 2.2 and Figure 3. The easiest way to prove that is to draw a line chart, in which the temperature improves as time goes. Further, samples with greater loading have a higher slope of temperature elevation. Based on my observation of data, this is a feasible solution.

Response: The graph that illustrates the temperature improvement and the accompanying description was added to the manuscript.

  • Please explain why results were provided selectively.
    • In figure 4, only AgNWs-DCF2 and AgNWs-SCF2 were used.
    • In figure 5, only AgNWs-DCF2 and AgNWs-SCF2 were used.

Response:

We considered the medium-sized sample of silver nano-wire coating as the optimal sample, which has both good heat reflection behavior and the average amount of nano-wire used in its construction (considering the high cost of nanowires, the final product may exceed expectations).

Is is noted that the tests performed at levels 1 and 3 of nanowire harvesting did not show a statistically significant difference in water vapor and air passage.

Minor Comments

  • Please ensure that all figures meet quality requirements as I can barely see the writing.

Response: The use of high-quality photos were included in the manuscript.

  • Line 241, “…blank sample there is no significant difference between them.
    • Significant differences must be supported through data analysis. A P-value needs to be provided.

Response: The mean data collected were shown from the difference between the blank sample and other samples and analyzed by one-way ANOVA using Tukey’s test (95% confidence intervals) by SSPSS V.18. A p-value <0.05 was considered statistically significant.

  • Line 306: “As shown in Figure 4 (1-5),…”.
    • I think the authors are referring to Figure 6.

Response: The corresponding figure now was corrected.

  • Table 1. Prepared samples with dip and spray methods
    • There are three “AgNWs-SCF1”

Response:  The description for the three samples was corrected. AgNWs-SCF1 for 0.1mg/cmof AgNWs , AgNWs-SCF2 for 0.3mg/cmof AgNWs and AgNWs-SCF3 for 0.5mg/cmof AgNWs.

In summary, the authors acknowledge the insightful and constructive comments of Reviewer #2. The corresponding changes are reflected in the revised manuscript. The authors have also comprehensively edited the manuscript for language, clarity, and syntax throughout to meet the high standards of this journal.

Reviewer 3 Report

Please find comments below and accept with minor revision: 

1) Please discuss mechanical testing of textiles coated with nanomaterials and thermal conductivity studies, Cite R. Villanueva et al.,  Materials Research Express 6, 016307 (2019). 

2) AgNWs coating when talked about dip-coating also cite: D. Ganta et al., Analytical Letters 53, 1992-2001 (2020).

3) Show curve for the effect of thickness on I-V measurements 

Author Response

Reviewer #3

Please find comments below and accept with minor revision: 

  • Please discuss mechanical testing of textiles coated with nanomaterials and thermal conductivity studies, Cite R. Villanueva et al.,  Materials Research Express 6, 016307 (2019).

Response:

The discussion about mechanical properties of coated textiles with this reference was added to the revised manuscript.

  • AgNWs coating when talked about dip-coating also cite: D. Ganta et al., Analytical Letters 53, 1992-2001 (2020).

Response:

The corresponding reference was added to the revised manuscript.

3) Show curve for the effect of thickness on I-V measurements 

Response:

Since the thickness of the coated nanowire is very small compared to the thickness of the substrate fabric, it was practically impossible to measure the thickness of the nanowire in the test, and our efforts to measure the thickness were not conclusive.

In summary, the authors acknowledge the insightful and constructive comments of Reviewer #3. The corresponding changes are reflected in the revised manuscript. As well, the authors have comprehensively edited the manuscript for language, clarity, and syntax throughout to meet the high standards of this journal.

Round 2

Reviewer 2 Report

The revised version of manuscript "Coating Cellulosic Material with Ag Nanowires to Fabricate Wearable IR-Reflective Device for Personal Thermal Management: The Role of Coating Method and Loading Level" has either addressed my concerns or provide answers why my concerns do not effect this research. I am convinced by the authors that the current version of manuscript is the best for this research and is ready for publication.